# Unraveling the Relationship between Sleep Problems, Emotional Behavior Disorders, and Stressful Life Events in Preschool Children

**DOI:** 10.3390/jcm11185419

**Published:** 2022-09-15

**Authors:** Filippo Manti, Federica Giovannone, Franca Aceti, Nicoletta Giacchetti, Francesca Fioriello, Andrea Maugeri, Carla Sogos

**Affiliations:** 1Department of Human Neuroscience, Unit of Child Neurology and Psychiatry, Sapienza University of Rome, 00185 Rome, Italy; 2Department of Human Neuroscience, Unit of Post-Partum Disorders, Sapienza University of Rome, 00185 Rome, Italy

**Keywords:** sleep problems, CSHQ, stressful life events, CLES-P, selective mutism, generalized anxiety disorder, oppositional defiant disorder

## Abstract

**Objectives**: The aims of this study were to: (1) explore sleep problems in preschool children with generalized anxiety disorder (GAD), selective mutism (SM), and oppositional defiant disorder (ODD) and (2) examine the relationship between stressful life events, sleep problems, and emotional behavior disorders in preschoolers. **Methods**: The parents of 213 preschool children with SM, GAD, ODD, and TD (typical development, age range 2–6 years) completed the Children’s Sleep Habits Questionnaire (CSHQ), the Coddington Life Events Scale, preschool version (CLES-P), and the CBCL 1½–5. **Results**: Eighty-three subjects reported sleep problems before the age of 2 years. Seventy-five children (86.14%) with a clinical diagnosis and eight children with TD (8.4%) exceeded the threshold level on the CSHQ. For the bedtime resistance (*p* = 0.042) and sleep duration subscales (*p* = 0.038), the SM group had significantly higher scores in comparison to the ODD group. The same pattern was also true for the sleep onset (*p* = 0.024) and sleep anxiety subscales (*p* = 0.019). The linear regression analysis model showed that the impact of stressful life events and internalizing problems could predict sleep habits in children. **Conclusions**: Emotional behavior disorders and stress factors should be regularly investigated in children who are referred to clinics for sleep problems. Clinicians should consider how these symptoms may exacerbate sleep problems and/or interfere with treatment.

## 1. Introduction

Sleep is an important issue in a child’s development, and sleep problems are predictive of long-term sleeping disorders, which may increase the risk of emotional and behavioral symptoms in young children [1,2,3]. The prevalence of childhood sleep disorders, such as difficulty falling asleep, difficulty staying asleep, or both, is estimated at 25% in the general pediatric population [4,5].

For many children, these problems represent transient difficulties that will remit without specific interventions. For others, sleep disorders can endure over an extended period of time and negatively affect the child’s development [6,7]. Compared to children without sleep problems, children with sleep problems tend to have more externalizing and internalizing symptoms [8,9,10]. In addition, children with sleep problems experience more difficulties with emotional and attentional regulation, socialization, and daily living skills [11].

There appears to be a complex, bidirectional relationship between sleep disorders and mental health. Sleep problems have been associated with epilepsy, autism spectrum disorder (ASD), attention deficit hyperactivity disorder (ADHD), mood and anxiety disorders, and a broad range of neurocognitive problems [12,13,14,15]. Furthermore, the presence of sleep disorders in preschoolers appears to predict a psychiatric condition during adolescence and adulthood, thereby emphasizing the need for sleep assessment as a precursor and risk factor for emotional behavior disorder onset [16,17,18]. Sleep problems may be one of the main reasons for parental stress in families with children with a psychiatric condition [19].

In a recent study, parents of children with ASD identified sleep problems as having a large impact on the child’s everyday life, emphasizing the importance of better understanding sleep patterns for these children [20]. However, only a few studies have focused on sleep problems and their association with emotional and behavioral problems and stressful life events in preschoolers [21].

The aims of this study were to: (1) explore the dimensions of sleep behavior in preschool children with generalized anxiety disorder (GAD), selective mutism (SM), and oppositional defiant disorder (ODD); (2) evaluate the relationship between sleep problems and internalizing and externalizing disorders in the clinical group as compared with age- and gender-matched controls; and (3) examine the relationship between stressful life events and sleep problems.

We hypothesized that sleep problems play an important role in the affect regulation process during childhood. Poor sleep in early childhood would be predictively associated with the onset of internalizing problems such as anxiety or selective mutism, and behavior problems, such as oppositional defiant disorders.

We predicted that sleep problems would be more common in children with an emotional and/or behavioral disorder than in controls and that there would be a specific sleep dimension problem in the different clinical conditions considered in this study. Furthermore, we hypothesized that sleep problems would be more common in emotional disorders than in behavioral disorders and that poor sleep may affect children’s self-regulation skills and consequently may interfere with behavioral expressions.

## 2. Materials and Methods

### 2.1. Subjects

The sample for the current study included 213 children between the ages of 2 and 6 years. Subjects were recruited from early intervention programs at the Department of Human Neuroscience of Sapienza University of Rome (n = 118; 55.4%) as well as from the general community (children with typical development (TD) = 0.95; 44.6%) in order to make comparisons between clinical and TD groups. 

Inclusion criteria for the clinical sample were: (1) a diagnosis of GAD, SM, or ODD; (2) age between 2 and 6 years; (3) medication free and not under any treatment for sleep problems at the time of assessment. Exclusion criteria for all samples were: (1) the occurrence of asthma, sleep apnea, or seizure; (2) comorbidity with any additional psychiatric disorders (ASD, ADHD, specific phobia, or separation anxiety disorder) and/or global developmental delay.

The clinical and TD groups were similar in terms of age, gender distribution, and sociodemographic characteristics (see Table 1).

### 2.2. Procedures

Clinical assessment of all children and their parents was performed according to a three-step process. The first step consisted of anamnesis and a clinical interview with the parents (demographic data, developmental stages, education, income, family history).

In the second step, children underwent a comprehensive neuropsychiatric evaluation (neuropsychological and psychopathological), during which the disorder was confirmed. Both parents were also asked to assess the emotional and behavioral problems of their child by completing a symptom checklist together (Child Behavior Checklist for Ages 1½–5 years, CBCL 1½–5) and the Coddington Life Events Scale, preschool version (CLES-P) to examine stressful life events.

In the third step, parents were asked to complete the Children’s Sleep Habits Questionnaire (CSHQ) in order to obtain further information on the child’s sleep/wake patterns.

The study was carried out in accordance with the Declaration of Helsinki and approved by the institutional review board of Sapienza University of Rome (Ref. 4434–prot. 354/17). All parents gave written informed consent to participate in this study.

### 2.3. Measures

#### 2.3.1. Background Information

Information on family history (demographic data, education, income, parental age, stressful life events) and developmental stages was collected on a separate short questionnaire. The information obtained from the anamnesis and clinical interviews with parents was subdivided into two domains:(1)Child characteristics:
(a)Sociodemographic variables: sex and age.(b)The presence/absence of sleep problems.(2)Family characteristics:
(a)Sociodemographic variables, such as parental age, nationality (considered immigrant when the family contained at least one immigrant parent), family structure (categorized in three groups: marriage/cohabitation, separation/divorce, or single), and income (socioeconomic status, evaluated by means of the Hollingshead scale) [22].(b)The presence/absence of family psychiatric status or history, depending on whether a mother, father, or other family member suffered from significant psychiatric health problems (we considered maternal, paternal, and family member psychiatric disorder responses).(c)Stressful life events, including the presence/absence of life stress, such as the loss of a significant person during infancy or adolescence, abuse during infancy or adolescence, familial and marital conflicts, complications during pregnancy and prematurity, physical health problems, and hospitalization, job loss, and home or country change (immigration).

#### 2.3.2. Emotional and Behavioral Assessment

Emotional and behavioral disorders were assessed according to the Diagnostic and Statistical Manual of Mental Disorders, Fifth Edition (DSM-5) criteria and confirmed by the Kiddie Schedule for Affective Disorders and Schizophrenia for School-Age Children- Present and Lifetime version (K-SADS-PL, Italian version) [23] and a complete clinical assessment.

Parental psychiatric diagnoses were made by a psychiatrist in the adult mental health service.

#### 2.3.3. Child Behavior Checklist for Ages 1½–5

The CBCL 1½–5 is a parent questionnaire that assesses emotional and behavioral problems in children. Parents rate their child on 100 items using a 3-point scale: 0 = not true, 1 = somewhat or sometimes true, and 2 = very true or often true. A total problem score, two broadband scores (internalizing and externalizing domains), and six different syndrome scales are used.

Six syndrome subscales contribute to either the broadband internalizing or externalizing domains. Only one syndrome scale, sleep problems, does not contribute to broadband but contributes to the total problems score. According to the psychometric information presented in the CBCL manual, the Sleep Problems scale has adequate internal consistency reliability (0.78) and test–retest reliability (0.92) [24]. 

For syndrome scales, the borderline clinical range spans from a T score of 65-69 (93rd to 97th percentile of a normative sample of non-referred children). Scores above the 97th percentile are considered to be in the clinical range. For internalizing, externalizing, and total scales, the borderline range spans from a T score of 60–63 (83rd to 90th percentile). The borderline DSM-oriented scale spans from a T score of 65–69 (93rd to 97th percentile of a normative sample of non-referred children).

### 2.4. Stress Measure

#### Coddington Life Events Scale (CLES)

The CLES is a validated and well-established questionnaire that measures the frequency and timing of both positive and negative life events during the last year (four trimesters). The CLES preschool version (CLES-P, Italian version), which contains 25 items, was used. By measuring significant life events (i.e., negative life events: *death of a parent; being hospitalized; marital separation of your parents, etc*.; positive life events: *major increase in your parents’ income; outstanding personal achievement; recognition for excelling in a sport or other activity, etc.*) in terms of life change units, the CLES-P can provide insight into recent events that may affect the child’s health. The CLES-P provides a life change unit score for each trimester and for the past 0–3, 0–6, 0–9, and 0–12 months. In addition to a total events score, a negative life events score is also calculated. Subjects with a score above the age-specific cut-off are considered to have a higher risk of developing psychological problems. The CLES has adequate internal consistency (0.75) and test–retest reliability (0.68) [25].

### 2.5. Sleep Instruments

#### Children’s Sleep Habits Questionnaire

The CSHQ is a parent-report sleep instrument that has been widely used to assess sleep disturbances in children. The questionnaire provides a total score and eight subscale scores: bedtime resistance, sleep onset delay, sleep duration, sleep anxiety, night wakings, parasomnias, sleep-disordered breathing, and daytime sleepiness. Parents rate the occurrence of 33 sleep habits or sleep problems as occurring “usually” (5–7 times/week), “sometimes” (2–4 times/week), or “rarely” (0–1 times/week) in the most recent typical week. Higher scores indicate more sleep problems. A total CSHQ score of 41 has been reported to be a sensitive clinical cut-off for the identification of probable sleep problems. The CSHQ has satisfactory internal consistency (0.81) and reliability (0.87) [26,27,28,29].

### 2.6. Data Analysis 

The Statistical Package for Social Sciences for Windows (SPSS) version 25.0 was used for all data analyses. Correlation analyses were estimated by Spearman’s correlation coefficient. Chi-square and Fisher’s exact tests were used to examine differences between groups for categorical data. For continuous data, independent sample t tests, one-way analysis of variance (ANOVA) with post hoc comparison (Bonferroni), or non-parametric tests (Kruskal–Wallis with post hoc Mann–Whitney U tests) were used. The dimensional effect of potential predictive clinical factors (CLES-P, CBCL scores) on the CSHQ composite score was evaluated by linear regression analysis. Statistical significance was set at *p* ≤ 0.05.

## 3. Results

Descriptive results of participant sociodemographic and clinical characteristics are presented in Table 1.

The clinical groups included 118 preschool children (GAD = 28 boys/16 girls, mean age of 53.5 months (SD: 10.1); SM = 11 boys/17 girls, mean age of 54.3 months (SD: 8.7); ODD = 30 boys/16 girls, mean age of 50.7 months (SD: 12.5)). A total of 95 children with TD were recruited through kindergarten screening (57 boys/38 girls, mean age of 47.13 months (SD: 10.11)).

Eighty-three subjects in the entire sample reported sleep problems before the age of 2 years. A total of 75 children (86.14%) with a clinical diagnosis (21 with SM, 29 with GAD, and 25 with ODD) and 8 children with TD (8.4%) exceeded the cut-off point on the CSHQ (score > 41), with significant differences between the TD and clinical groups (*p* < 0.001, see Table 2).

CSHQ subscale scores were compared between the three clinical groups using ANOVA. Statistically significant differences were found on four subscales. For the bedtime resistance subscale (F(2, 115) = 10.39, *p* = 0.042), the group with SM had significantly higher scores in comparison to the ODD group. The same pattern was also true for sleep onset (F(2, 115) = 0.70, *p* = 0.024) and sleep anxiety subscales (F(2, 115) = 5.59, *p* = 0.019). The SM group had higher scores on the sleep duration subscale (F(2, 115) = 3.22, *p* = 0.038) than the other two clinical groups. 

Significant differences in CBCL 1½−5 scores were found between the groups (see Table 3). In the SM group, a significant correlation was found between the CSHQ night waking subscale and the CBCL aggressive subscale (r = 0.420, *p* = 0.026) and between the CSHQ total score and the CBCL aggressive subscale (r = 0.389, *p* = 0.041). In the GAD group, significant correlations were found between the CSHQ sleep-disordered breathing subscale and the CBCL sleep problems (r = 0.323, *p* = 0.033), CBCL total problems (r = 0.300, *p* = 0.048), and CBCL affective subscales (r = 0.316, *p* = 0.036).

Regarding the parental perception of their child’s psychopathology, children with ODD received higher scores than other groups on the CBCL total and externalizing scales. Parents of children with ODD reported higher scores than those of children with GAD or SM on the affective (*p* = 0.036), ADHD (*p* = 0.004), and oppositional defiant problems subscales (*p* = 0.002). In addition, these parents reported higher scores than those of children with SM on ADHD (*p* = 0.000) and oppositional defiant problems subscales (*p* = 0.002). No significant correlations were found between CBCL subscales and CSHQ scores in the ODD group.

On the CBCL subscales, parents of children with GAD and/or SM reported higher scores than parents of children with TD on the sleep problems (GAD vs. TD, *p* = 0.001; SM vs. TD, *p* = 0.001), affective problems (GAD vs. TD, *p* = 0.001; SM vs. TD, *p* = 0.012), anxiety problems (GAD vs. TD, *p* = 0.000; SM vs. TD, *p* = 0.001), and pervasive developmental problems subscales (GAD vs. TD, *p* = 0.001; SM vs. TD, *p* = 0.000). As regards family characteristics, significant differences were found between the clinical and TD groups (see Table 1).

CLES-P subscale scores were compared among the three clinical groups using ANOVA. For the CLES-P 0–3 subscale (F(2, 115) = 6.97, *p* = 0.001), the SM group had significantly higher scores in comparison to the GAD and ODD groups. The same pattern was also true for CLES-P 0–6 (F(2, 115) = 5.20, *p* = 0.007) and CLES-P 0–9 subscale scores (F(2, 115) = 4.65, *p* = 0.011). The SM group had higher CLES-P 0–12 subscale scores (F(2, 115) = 4.29, *p* = 0.016) than the other two clinical groups (see Table 4). 

As regards the relationship between the CBCL 1½–5, CSHQ, and, CLES-P scales see Table 5.

Finally, significant correlations emerged between the CBCL 1½–5 and CLES-P subscales in the GAD group: sleep problems vs. CLES-P 0–3 subscale (r = 0.299; *p* = 0.048); sleep problems vs. CLES-P 0–9 subscale (r = 0.361, *p* = 0.016); and sleep problems vs. CLES-P 0–12 subscale (r = 0.409, *p* = 0.006).

No significant correlations between the CSHQ, CBCL 1½–5, and CLES-P subscales were found in the ODD group.

A regression analysis was performed in order to identify clinical variables that best predicted sleep problems in the clinical sample. A stepwise method was applied with the CSHQ score as the dependent variable and CLES-P 0–3, CLES-P 0–6, CLES-P 0–9, CLES-P 0–12, and CBCL internalizing, externalizing, and total scale scores as independent variables. We found a significant influence of the CBCL internalizing (*p* = 0.000) and CLES-P 0–3 subscales (*p* = 0.000) on CSHQ score variability, and this model explained 21% of the variance (R^2^ = 0.210).

We applied the same fixed model in the three different clinical groups. In the SM group, we found a significant influence of the CLES-P 0–9 subscale (*p* = 0.024) on CSHQ score variability, and this model explained 18% of the variance (R^2^ = 0.182).

## 4. Discussion

There is considerable evidence that inadequate sleep can adversely affect cognitive, emotional, and behavioral functioning. For this reason, screening for and assessing sleep problems in preschool children with an emotional and behavioral disorder is critical to their care [4].

To our knowledge, a direct comparison between sleep problems and stressful life events in preschool children with SM, GAD, and ODD has not been conducted. The prevalence of sleep problems in our clinical group (86.14%) was higher than would be expected based on previous research in preschool children [4,6,8]. Bedtime resistance and sleep duration were more prevalent in the group with SM than in the other two clinical groups. Sleep anxiety was most significantly associated with the SM and GAD groups and sleep onset delay was often the reason that children with GAD were referred for clinical treatment.

In our study, children with internalizing problems (SM and GAD groups) showed more sleep problems than children with externalizing problems (ODD group). Consistent with previous studies [30,31], we found that sleep problems, especially those involving sleep onset delay and sleep duration, were significantly associated with emotional and behavioral problems.

Childhood is a critical time when children develop the ability to regulate their emotions. Sleep problems, if untreated, may be persistent, have adverse effects on child development, and may predict the persistence of emotional and behavioral symptoms across childhood and adulthood [32,33,34,35]. Previous studies have found links between sleep problems and several psychiatric and neurological disorders [36].

We also investigated the association between child sleep habits and emotional behavioral problems using the CSHQ questionnaires and CBCL. Our results showed that co-sleeping and night wakings were associated with poorer sleep quality in the SM group. We hypothesize that sleeping in the parents’ room or bed may be a rewarding experience that enhances sleep fragmentation and child irritability. Concerning the GAD group, we hypothesize that bedtime resistance and sleep onset delay are associated with separation from caregivers at bedtime. This condition could be related to a reduced sense of security and comfort in children who have more difficulty with self-soothing. Concerning the ODD group, children showed difficulty with sleep duration but more preserved self-soothing in comparison to the other two clinical groups. Parents of children with ODD reported an increased vulnerability for ADHD symptoms and sleep problems on the CBCL scales. However, it should be noted that the CBCL ADHD scale does not capture all ADHD symptoms and does not separate inattentive and hyperactive-impulsive symptoms, which are differentially related to sleep problems [37]. As a previous study [38], daytime sleepiness in children with ODD was often reported by parents such as increased impulsivity, activity levels, acting out, and inattention.

Many authors [39,40] have suggested that early life experiences of adversity or parental psychological disorders could have a negative impact on emotional functioning. The present study reported the association between stressful life events, emotional and behavioral disorders, and sleep problems, an association that has been insufficiently investigated in preschool children. Our results may provide potential insights that can inform a clinical index of suspicion regarding undetected emotional and behavioral problems in the presence of specific sleep patterns or vice versa.

The proportion of children with critical CLES-P scores was consistently higher in the SM group compared with the two other clinical groups, showing that stress and impairment in the regulation and processing of emotions may be more significant in the SM group. Capozzi et al. [41] confirmed a relationship between being a parent of a child with SM and traumatic experiences. They hypothesized that parents of children with SM were unable to identify the severity of trauma and the consequences it might have on their emotional functioning and family interactions. Furthermore, significant correlations were found between CBCL internalizing and sleep problem scales and higher stress scores, suggesting that emotional dysregulation in the SM group could result in a reduced ability to cope with stress and increased vulnerability to sleep problems.

The number of stressful life events (such as a family member’s death, divorce, recent change of residence, or physical problems) in the previous year could predict a child’s sleep and emotional and behavioral problems. The linear regression analysis model showed that the impact of stressful life events and internalizing problems could predict sleep problems in children with an emotional and behavioral condition.

Finally, the CSHQ appears to be a useful sleep screening instrument to delineate sleep habits and identify problematic sleep domains in preschool children with emotional and behavioral disorders. 

Emotional and behavioral disorders and stress factors should be regularly investigated in children referred to clinics for sleep problems. Clinicians should consider how these symptoms may exacerbate sleep problems and/or interfere with treatment. Increased levels of parental stress could further highlight the importance of assessing and treating sleep problems in children with psychiatric disorders.

The high prevalence of sleep problems identified in the current study, which is comparable to that reported in other community samples of preschool-aged children [42,43], confirms the pervasiveness of sleep problems and highlights the importance of understanding the factors that cause and maintain them. 

We would like to conclude with a particular emphasis on high prevalence of sleep problems and life events in SM. Since SM is a very challenging and complex disorder that needs a combination of treatment, every bit of new data is extremely important and valuable.

Based on our results, we feel that ongoing sleep complaints in early childhood should be considered a “red flag” for further assessment of psychiatric disorders. Knowledge of the relative frequency and specific types of sleep problems in clinical populations is important for both clinical and research purposes, including the development and implementation of targeted interventions.

## 5. Limitations

One limitation concerned the sleep measure used in this study. The generalizability of this study’s findings should be evaluated using other measures of sleep problems, such as actigraphs. Longitudinal data could confirm the early presentation of chronic sleep problems as a prognostic indicator of emotional and behavioral problems in adolescence and adulthood. 

## Figures and Tables

**Table 1 jcm-11-05419-t001:** Clinical and demographic characteristics of the preschool sample.

	SM Group (n = 28) M (SD)/Freq (%)	GAD Group (n = 44) M (SD)/Freq (%)	ODD Group (n = 46) M (SD)/Freq (%)	TD Group (n = 95) M (SD)/Freq (%)	*p*
Child age (months)	54.01 (8.57)	53.64 (10.22)	50.67 (12.54)	47.13 (10.1)	-
**Sex**					
Male	11 (39%)	28 (64%)	30 (65%)	57 (60%)	-
Female	17 (61%)	16 (36%)	16 (35%)	38 (40%)	-
Mother age	37.00 (5.13)	35.93 (4.60)	37.00 (5.13)	33.68 (5.82)	0.003 **
Father age (yy/mm)	34.89 (4.63)	38.61 (5.96)	39.98 (6.22)	36.63 (5.82)	0.002 **
**Family structure**					
Nuclear family	26 (92.8%)	42 (95.4%)	43 (93.4%)	88 (92.7%)	-
Extended family	1 (3.6%)	2 (5.6%)	3 (6.6%)	6 (6.3%)	-
Single parent	1 (3.6%)	-	-	1 (1%)	-
**Nationality**					
Caucasian	27 (96.4%)	44 (100%)	42 (91.3%)	90 (94.7%)	-
Non-Caucasian	1 (3.6%)	-	4 (8.7%)	5 (5.3%)	-
**Psychiatric status**					
Maternal psychiatric disorders	5 (17.8%)	7 (15.9%)	5 (10.8%)	1 (1%)	0.004 **
Paternal psychiatric disorders	5 (17.8%)	8 (18.1%)	4 (8.7%)	1 (1%)	0.002 **
Psychiatric disorders in family member	3 (10.7%)	3 (6.8%)	2 (4.3%)	1 (1%)	-
Stressful life events (last 3 months)	8 (28.6%)	10 (22.7%)	12 (26.1%)	7 (7.4%)	0.006 **
Stressful life events (last 12 months)	20 (71.4%)	38 (86.3%)	27 (58.7%)	7 (7.4%)	0.000 **
**Income**					
Below EUR 30,000 per year	5 (17.8%)	3 (6.8%)	5 (10.8%)	7 (7.4%)	-
EUR 30,000–60,000 per year	21 (75%)	34 (77.2%)	36 (78.2%)	75 (78.9%)	-
Above EUR 60,000 per year	2 (7.1%)	7 (15.9%)	5 (10.8%)	13 (13.7%)	-
Subjects with a sleep disorder (n)	22 (78.6%)	28 (63.6%)	25 (54.3%)	8 (8.42%)	0.000 **

Legend: M—mean; SD—standard deviation; SM—selective mutism; ODD—oppositional defiant disorder; GAD—generalized anxiety disorder; TD—typical development. ** *p* < 0.010.

**Table 2 jcm-11-05419-t002:** Between-group comparisons of the Children’s Sleep Habit Questionnaire.

CSHQ	SM Group M (SD) n = 28	GAD Group M (SD) n = 44	ODD Group M (SD) n = 46	TD Group M (SD) n = 95	Post Hoc	*p*
**Bedtime Resistance**	10.68 (2.67)	10.27 (3.26)	8.91 (3.48)	6.58 (1.38)	SM vs. GAD	ns
					SM vs. ODD	0.017
					GAD vs. ODD	0.050
					SM + GAD + ODD vs. TD	**0.000 ****
**Sleep Onset Delay**	1.89 (8.75)	1.93 (0.92)	1.48 (0.72)	1.04 (0.25)	SM vs. GAD	ns
					SM vs. ODD	0.040
					GAD vs. ODD	0.012
					SM + GAD + ODD vs. TD	**0.000 ****
**Sleep Duration**	5.46 (2.20)	4.43 (1.60)	4.48 (1.69)	3.16 (0.41)	SM vs. GAD	0.037
					SM vs. ODD	0.048
					GAD vs. ODD	ns
					SM + GAD + ODD vs. TD	**0.000 ****
**Sleep Anxiety**	7.43 (1.97)	6.88 (2.57)	5.89 (2.37)	4.33 (0.98)	SM vs. GAD	ns
					SM vs. ODD	**0.004 ****
					GAD vs. ODD	ns
					SM + GAD + ODD vs. TD	**0.000 ****
**Night Wakings**	4.64 (1.73)	4.59 (1.78)	4.30 (1.60)	3.14 (0.54)	SM vs. GAD	ns
					SM vs. ODD	ns
					GAD vs. ODD	ns
					SM + GAD + ODD vs. TD	**0.000 ****
**Parasomnias**	8.93 (1.05)	8.86 (1.96)	8.83 (1.97)	7.20 (0.75)	SM vs. GAD	ns
					SM vs. ODD	ns
					GAD vs. ODD	ns
					SM + GAD + ODD vs. TD	**0.000 ****
**Sleep Disordered Breathing**	3.14 (0.76)	3.14 (0.35)	3.48 (1.22)	3.12 (0.38)	SM vs. GAD	ns
					SM vs. ODD	ns
					GAD vs. ODD	ns
					SM + GAD + ODD vs. TD	**0.000 ****
**Daytime Sleepiness**	11.43 (2.35)	11.03 (2.64)	11.74 (3.04)	11.01 (2.80)	SM vs. GAD	ns
					SM vs. ODD	ns
					GAD vs. ODD	ns
					SM + GAD + ODD vs. TD	**0.000 ****
**Total CSHQ Score**	49.68 (7.50)	47.23 (8.59)	46.00 (10.00)	37.32 (3.89)	SM vs. GAD	ns
					SM vs. ODD	0.076
					GAD vs. ODD	ns
					SM + GAD + ODD vs. TD	**0.000 ****
**Sleep Duration on Weekday** **(min)**	620.35 (46.23)	624.55 (62.82)	609.00 (77.32)	642.00 (56.57)	SM vs. GAD	ns
					SM vs. ODD	ns
					GAD vs. ODD	ns
					SM+GAD+ODD vs. TD	**0.000 ****
**Sleep Duration on Weekend** **(min)**	643.93 (68.08)	636.82 (74.29)	614.02 (72.83)	707.21 (78.40)	SM vs. GAD	ns
					SM vs. ODD	ns
					GAD vs. ODD	ns
					SM + GAD + ODD vs. TD	**0.000 ****

Legend: CSHQ—Children’s Sleep Habits Questionnaire; M—mean; SD—standard deviation; SM—selective mutism; ODD—oppositional defiant disorder; GAD—generalized anxiety disorder; TD—typical development. Significant differences after Bonferroni correction are marked in bold. ** *p* < 0.01; ns = not significant.

**Table 3 jcm-11-05419-t003:** Results of analysis of variance of Child Behavior Checklist subscale scores in the preschool sample (n = 213).

CBCL 1½–5 (T-Score)	SM Group M (SD) n = 28	GAD Group M (SD) n = 44	ODD Group M (SD) n = 46	TD Group M (SD) n = 95	*p*
**Internalizing Problems**	62.29 (8.30)	59.14 (10.27)	62.74 (10.97)	47.15 (9.84)	**0.000 ****
**Externalizing Problems**	51.78 (7.86)	54.25 (9.08)	61.06 (10.06)	48.38 (6.92)	**0.000 ****
**Total Problems**	57.50 (8.55)	57.75 (8.95)	63.13 (9.53)	48.38 (7.35)	**0.000 ****
**Affective Problems**	57.50 (9.08)	57.27 (8.04)	61.20 (9.44)	52.77 (3.58)	**0.000 ****
**Anxiety Problems**	61.96 (12.90)	59.50 (9.95)	60.08 (6.85)	52.29 (5.72)	**0.000 ****
**Pervasive Developmental Problems**	65.89 (9.40)	61.55 (8.91)	63.41 (9.51)	56.35 (6.41)	**0.000 ****
**Attention Deficit Hyperactivity Problems**	53.64 (4.67)	56.43 (6.59)	61.02 (7.88)	54.86 (4.48)	**0.000 ****
**Oppositional Defiant Problems**	54.00 (6.12)	53.36 (9.57)	59.59 (8.69)	51.58 (3.23)	**0.000 ****
**Sleep Problems Subscale**	57.55 (7.17)	56.70 (8.39)	58.11 (8.40)	52.06 (4.83)	**0.000 ****

Legend: CBCL 1½–5—Child Behavior Checklist for Ages 1½–5; SD—standard deviation; SM—selective mutism; GAD—generalized anxiety disorder; ODD—oppositional defiant disorder; TD—typical development. Significant differences after Bonferroni correction are marked in bold. ** *p* < 0.01.

**Table 4 jcm-11-05419-t004:** Between-group comparisons of the Coddington Life Events Scale Preschool Version.

CLES-P Negative Event Score	SM Group M (SD) n = 28	GAD Group M (SD) n = 44	ODD Group M (SD) n = 46	TD Group M (SD) n = 95	Post Hoc	*p*
**Last 3 months**	72.46 (93.21)	20.73 (32.97)	27.98 (54.74)	2.21 (9.43)	SM vs. GAD	0.008
					SM vs. ODD	0.027
					GAD vs. ODD	ns
					SM + GAD + ODD vs. TD	**0.000 ****
**Last 6 months**	74.96 (90.81)	25.18 (46.06)	34.13 (64.74)	2.61 (10.11)	SM vs. GAD	0.011
					SM vs. ODD	0.043
					GAD vs. ODD	ns
					SM + GAD + ODD vs. TD	**0.000 ****
**Last 9 months**	77.57 (93.06)	27.79 (49.38)	37.50 (69.48)	10.49 (1.08)	SM vs. GAD	0.013
					SM vs. ODD	0.055
					GAD vs. ODD	ns
					SM + GAD + ODD vs. TD	**0.000 ****
**Last 12 months**	78.00 (92.33)	29.18 (50.48)	39.26 (72.37)	3.99 (12.25)	SM vs. GAD	0.014
					SM vs. ODD	0.064
					GAD vs. ODD	ns
					SM + GAD + ODD vs. TD	**0.000 ****

Legend: CLES-P—Coddington Life Events Scale-Preschool; M—mean; SD—standard deviation; SM—selective mutism; ODD—oppositional defiant disorder; GAD—generalized anxiety disorder; TD—typical development. Significant differences after Bonferroni correction are marked in bold. ** *p* < 0.01; ns = not significant.

**Table 5 jcm-11-05419-t005:** Spearman *r* correlations between clinical variables in the group with selective mutism.

	CLES-P (0–3)	CLES-P (0–6)	CLES-P (0–9)	CLES-P (0–12)
**CBCL 1 ½–5** **Syndrome Scale**				
Sleep Problems	*r* = 0.424 *p* = 0.025 *	*r* = 0.477 *p* = 0.010 **	*r* = 0.526 *p* = 0.004 **	*r* = 0.545 *p* = 0.003 **
**Internalizing Problems**	ns	*r* = 0.433 *p* = 0.021 *	*r* = 0.480 *p* = 0.010 *	*r* = 0.477 *p* = 0.010 *
**Externalizing Problems**	ns	ns	*r* = 0.437 *p* = 0.020 *	*r* = 0.440 *p* = 0.019 *
**Total Problems**	ns	*r* = 0.404 *p* = 0.033 *	*r* = 0.475 *p* = 0.011 *	*r* = 0.440 *p* = 0.019 *
**CSHQ**				
Bedtime Resistance	ns	ns	ns	ns
Sleep Onset Delay	ns	ns	ns	ns
Sleep Duration	ns	ns	ns	ns
Sleep Anxiety	ns	*r* = 0.440 *p* = 0.019 *	*r* = 0.435 *p* = 0.021 *	*r* = 0.445 *p* = 0.018 *
Night Wakings	ns	ns	*r* = 0.385 *p* = 0.043 *	*r* = 0.385 *p* = 0.043 *
Parasomnias	ns	ns	ns	ns
Sleep Disordered Breathing	ns	ns	ns	ns
Daytime Sleepiness	ns	ns	ns	ns
**Total CSHQ score**	ns	*r* = 0.376 *p* = 0.049 *	*r* = 0.407 *p* = 0.032 *	*r* = 0.406 *p* = 0.032 *

Legend: CLES-P—Coddington Life Events Scale Preschool. * *p* < 0.05; ** *p* < 0.01; ns = not significant.

## Data Availability

The data presented in this study are available on request from the corresponding author.

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
