# Peer review of "Unraveling the Relationship between Sleep Problems, Emotional Behavior Disorders, and Stressful Life Events in Preschool Children"

_jcm, 2022, doi:10.3390/jcm11185419_

Round 1

Reviewer 1 Report

The paper addresses a problem in need of research. On the one hand, sleep problems are known to be part of a broader clinical syndrome of behavioral and emotional disorders in many children, particularly young children. On the other hand, it is important to differentiate between behavioral and emotional disorders in children with the highest potency of sleep difficulties. I missed this emphasis in the introduction to the article. I think the authors should more clearly articulate the purpose of their work and the rationale for the research questions they pose.
The methodology of the study is appropriate, but the psychometric potential of the methods used, especially the validity, should be stated. I believe that the CBCL Sleep difficulties subscale and the CSHQ may have overlap in the assessment of sleep problems, but the authors do not reflect this either in the presentation of the results or in the discussion.
For me, the interesting result of this study is that the group of children with SM has the highest scores for problematic sleep habits, and that sleep difficulties here are predicted by stressful life events. I think the authors should pay more attention to highlighting the most important findings of their work. This would allow for a more focused discussion.

Author Response

Reviewer #1

The paper addresses a problem in need of research. On the one hand, sleep problems are known to be part of a broader clinical syndrome of behavioral and emotional disorders in many children, particularly young children. On the other hand, it is important to differentiate between behavioral and emotional disorders in children with the highest potency of sleep difficulties. I missed this emphasis in the introduction to the article. I think the authors should more clearly articulate the purpose of their work and the rationale for the research questions they pose.

Authors’ reaction: we agree with the Reviewer’s comments. The manuscript has been modified accordingly.

The methodology of the study is appropriate, but the psychometric potential of the methods used, especially the validity, should be stated. I believe that the CBCL Sleep difficulties subscale and the CSHQ may have overlap in the assessment of sleep problems, but the authors do not reflect this either in the presentation of the results or in the discussion.

Authors’ reaction: We would like to thank the reviewer for comments and suggestions. Regarding possible overlapping of CBCL sleep subscale with CSHQ, we found that CBCL provides a photograph of whether common sleep problems occur and it is a useful tool as a potential screening measure. On the other hand, CSHQ is a sleep-specific measure that studies more in-depth sleep habits in youth and has the potential to accurately support clinicians in the diagnostic process.

For me, the interesting result of this study is that the group of children with SM has the highest scores for problematic sleep habits, and that sleep difficulties here are predicted by stressful life events. I think the authors should pay more attention to highlighting the most important findings of their work. This would allow for a more focused discussion.

Authors’ reaction: We would like to thank the reviewer for comments and suggestions. The manuscript has been modified accordingly.

Reviewer 2 Report

This was an interesting paper with both clinical and conceptual implications. The rationale is straightforward, as was the presentation. I applaud the authors for using non-parametric statistics on data that are not guaranteed to provide interval-level measurements. That said, there are some opportunities to clarify the material/descriptions and thus strengthen the presentation. Specific comments follow.

1. Please provide internal reliabilities for all the measures in the present sample. Also, it might be helpful for the authors to justify or otherwise explain why they selected the instruments they did. One particular point related to the CLES is whether it considers only adverse life events (distress) or also positive stress (eustress events)? Even positive stress can affect one's physiology and psychology, so I'd be curious to understand what the tool means by "significant life events."  Sample items from each measure would help here.

2. Clarify whether Bonferroni corrections were applied only with post hoc analyses or used throughout. I would expect, as exploratory data, that corrections for multiple observations are needed as there do not seem to be precisely planned hypotheses being tested, only general ones. From my calculations, a Bonferroni correction (.05) applied across all analyses in the Tables would stipulate statistical significance at .0007 or lower. It's unclear how this would effect the various effects reported here.

3. Please provide a correlation matrix for all the measures used rather than give those in text. Of course, any broad Bonferroni correction should also take into account these correlations as well, and so the larger n would lower the alpha level of .0007 above.

Author Response

Reviewer #2

This was an interesting paper with both clinical and conceptual implications. The rationale is straightforward, as was the presentation. I applaud the authors for using non-parametric statistics on data that are not guaranteed to provide interval-level measurements. That said, there are some opportunities to clarify the material/descriptions and thus strengthen the presentation. Specific comments follow.

  1. Please provide internal reliabilities for all the measures in the present sample. Also, it might be helpful for the authors to justify or otherwise explain why they selected the instruments they did. One particular point related to the CLES is whether it considers only adverse life events (distress) or also positive stress (eustress events)? Even positive stress can affect one's physiology and psychology, so I'd be curious to understand what the tool means by "significant life events." Sample items from each measure would help here.

Authors’ reaction: we agree with the Reviewer’s comments. The manuscript has been modified accordingly.

  1. Clarify whether Bonferroni corrections were applied only with post hoc analyses or used throughout. I would expect, as exploratory data, that corrections for multiple observations are needed as there do not seem to be precisely planned hypotheses being tested, only general ones. From my calculations, a Bonferroni correction (.05) applied across all analyses in the Tables would stipulate statistical significance at .0007 or lower. It's unclear how this would effect the various effects reported here.

Authors’ reaction: we thank the reviewer for giving us the possibility to clarify this point.

Bonferroni corrections were applied only with post hoc analyses. The tables were revised according to the reviewer’s request.

  1. Please provide a correlation matrix for all the measures used rather than give those in text. Of course, any broad Bonferroni correction should also take into account these correlations as well, and so the larger n would lower the alpha level of .0007 above.

Authors’ reaction: Table 5 has been added to the manuscript.